# Dalbavancin, Vancomycin and Daptomycin Alone and in Combination with Cefazolin against Resistant Phenotypes of *Staphylococcus aureus* in a Pharmacokinetic/Pharmacodynamic Model

**DOI:** 10.3390/antibiotics9100696

**Published:** 2020-10-14

**Authors:** Jacinda C. Abdul-Mutakabbir, Razieh Kebriaei, Kyle C. Stamper, Zain Sheikh, Philip T. Maassen, Katherine L. Lev, Michael J. Rybak

**Affiliations:** 1Department of Pharmacy Practice, Eugene Applebaum College of Pharmacy and Health Sciences, Wayne State University, Detroit, MI 48201, USA; jabdulmutakabbir@llu.edu (J.C.A.-M.); r.kebriaei@gmail.com (R.K.); kstamper@wayne.edu (K.C.S.); Zain.sheikh@wayne.edu (Z.S.); philip.maassen@wayne.edu (P.T.M.); katlev@wayne.edu (K.L.L.); 2School of Medicine, Wayne State University, Detroit, MI 48201, USA

**Keywords:** *Staphylococcus aureus*, antimicrobial resistance, pharmacokinetics, pharmacodynamics

## Abstract

The most efficacious antimicrobial therapy to aid in the successful elimination of resistant *S. aureus* infections is unknown. In this study, we evaluated varying phenotypes of *S. aureus* against dalbavancin (DAL), vancomycin (VAN), and daptomycin (DAP) alone and in combination with cefazolin (CFZ). The objective of this study was to observe whether there was a therapeutic improvement in adding a beta-lactam to a glycopeptide, lipopeptide, or a lipoglycopeptide. We completed a series of in vitro tests including minimum inhibitory concentration testing (MIC) of the antimicrobials in combination, time-kill analysis (TKA), and a 168 h (7-day) one-compartment pharmacokinetic/pharmacodynamic (PK/PD) model on two daptomycin non-susceptible (DNS), vancomycin intermediate *S. aureus* strains (VISA), D712 and 6913. Results from our MIC testing demonstrated a minimum 2-fold and a maximum 32-fold reduction in MIC values for DAL, VAN, and DAP in combination with CFZ, in contrast to either agent used alone. The TKAs completed on four strains paralleled the enhanced activity demonstrated via the combination MICs. In the one-compartment PK/PD models, the combination of DAP plus CFZ or VAN plus CFZ resulted in a significant (*p* < 0.001) improvement in bactericidal activity and overall reduction in CFU/ml over the 7-day period. While the addition of CFZ to DAL improved time to bactericidal activity, DAL alone demonstrated equal and more sustained overall activity compared to all other treatments. The use of DAL alone, with or without CFZ and the combinations of VAN or DAP with CFZ appear to result in increased bactericidal activity against various recalcitrant *S. aureus* phenotypes.

## 1. Introduction

The appropriate management of methicillin-resistant *Staphylococcus aureus* (MRSA) infections has become increasingly complex due to the limited efficacy of glycopeptides in bacterial eradication, commonly attributed to the emergence of drug-resistant strains [1,2,3]. Daptomycin (DAP), a bactericidal lipopeptide, has been shown to be a viable alternative amid vancomycin (VAN) resistance; nevertheless, increased VAN minimum inhibitory concentrations (MICs) against *S. aureus* have been shown to parallel with increased DAP MICs [4,5,6]. The currently recommended alternatives in lieu of elevated VAN and DAP MICs are minimal due to the associated undesirable side effects, bacteriostatic activity, and the achievement of suboptimal antimicrobial concentrations [7,8]. There is a critical need for safe and efficacious antimicrobial therapies to manage tolerant/resistant *S. aureus* infections.

Dalbavancin (DAL) is a long-acting lipoglycopeptide that has demonstrated enhanced activity eliminating highly resistant *S. aureus* [9]. This antibiotic is approved by the FDA for the management of acute bacterial skin and skin structure infections (ABSSSIs). DAL is best characterized by its long lipophilic chain responsible for its enhanced half-life of 147 to 258 h, which allows for a single-dose administration [10]. More importantly, DAL has been shown to be more potent, in vitro, than glycopeptides such as VAN and teicoplanin in the eradication of multidrug-resistant (MDR) Gram-positive bacteria [9,10,11].

Despite the enhanced activity observed with DAL in comparison to VAN and DAP, it is important to consider alternative therapeutic options to enhance infection elimination and to reduce the propensity of *S. aureus* to developing resistance [12]. The use of VAN or DAP in combination with anti-staphylococcal beta-lactams has been shown to be synergistic and bactericidal against MRSA, while also preventing the emergence of vancomycin intermediate resistant (VISA) and daptomycin non-susceptible (DNS) strains [13,14,15,16]. The objective of this study was to evaluate DAL, VAN, and DAP alone and in combination with cefazolin (CFZ) against varying phenotypes of *S. aureus*, including DNS and DNS-VISA strains, through a series of in-vitro testing experiments, including: single minimum inhibitory concentration (MIC) testing, combination MIC testing, time-kill analysis (TKA), and one-compartment in-vitro pharmacokinetic/pharmacodynamic (PK/PD) modelling.

## 2. Results

### 2.1. Susceptibility Testing

The susceptibility results of the single MICs and the combination MICs can be found in Table 1. DAL or DAP in combination with CFZ showed an average of an 8-fold reduction in DAL and DAP monotherapy MIC values in the MRSA, hVISA, and VISA phenotypes, while, in the DNS and DNS-VISA phenotypes, the DAL or DAP plus CFZ combination therapies resulted in a 32-fold and 16-fold decline in DAL and DAP monotherapy MIC values, respectively. In the VAN plus CFZ combinations, a 16-fold reduction in the DNS and DNS-VISA MIC values and an average 8-fold reduction in the remaining phenotypes were observed. Overall, the reduction in MIC values seen in the combination therapies was significantly greater when compared to the monotherapy values (*p* < 0.05).

### 2.2. In Vitro Time-Kill Analysis

All combinations demonstrated both synergistic and bactericidal activity. In the TKAs, the DAL and VAN plus CFZ combination therapies both demonstrated an average 4.95 log_10_ CFU/mL bacterial reduction from the most active single agent in each organism observed; while the DAP plus CFZ combination therapy showed an average 4.63 log_10_ CFU/mL reduction from the most active single agent. The results of the 4 TKAs are shown in Figure 1.

### 2.3. In Vitro One-Compartment PK/PD Model

#### 2.3.1. Pharmacokinetics

The PK parameters observed were within 2%, 1.5%, 5%, and 11% of the targeted values for DAL, VAN, DAP and CFZ, respectively. The attained and targeted PK parameters as well as the *f*AUC/MIC values for each treatment regimen are shown in Table 2.

#### 2.3.2. Pharmacodynamics

A starting inoculum of approximately 10^7^ was achieved in the one-compartment PK/PD model. Against both the DNS-VISA strains evaluated in the PK/PD model, D712 and 6913, the DAP and VAN monotherapies demonstrated bacteriostatic activity that was not sustained with either antimicrobial, with regrowth being observed at the 48h time points for both DAP and VAN. In contrast the DAL monotherapy maintained its bactericidal activity throughout the model duration, in each of the tested strains. The combination of DAP plus CFZ demonstrated rapid and consistent bactericidal activity, beginning at the 4h timepoint and continuing for the duration of the 168 h model, in both D712 and 6913. Despite small bacterial growth increases (<1-log CFU/mL) observed throughout the model conducted in each strain, the DAP plus CFZ combination reached detection limits by the end of treatment. The VAN plus CFZ combination demonstrated bactericidal activity against each of the strains. Nonetheless, regrowth was observed at the 96 h time point and the 48 h time points, in 6913 and D712, respectively; with that regrowth sustained throughout the completion of each model. Despite the regrowth observed in the VAN plus CFZ model for either strain evaluated, the combination model maintained a >2-fold CFU/mL reduction in comparison to the VAN monotherapy. The addition of CFZ to DAL did present with an improved time to bactericidal activity noted by a >3-log CFU/mL reduction from the initial inoculum at 8 h rather than in 24 h as shown in the DAL monotherapy; however, this was not statistically significant. For both the D712 and 6913 DNS-VISA strains, the combinations of DAP or VAN plus CFZ demonstrated significant log_10_ CFU/mL reductions in comparison to the DAP and VAN monotherapies (*p* < 0.001). Nevertheless, the DAL monotherapy presented with equal and in some instances more sustained activity than DAL, VAN, or DAP plus CFZ. No resistant mutants were recovered from any of the models with either organism at the 168 h conclusion of the models. A VAN Population Analysis Profile (PAP) of the parent strains, D712 and 6913, displayed a PAP/AUC ratio of 1.2 to Mu3, thus indicating the presence of VAN-resistant subpopulations. An additional PAP analysis of the organisms, recovered at the 168 h time point of the VAN+CFZ models, once again revealed a PAP/AUC ratio of 1.2 when compared to Mu3, signifying the absence of a shift or an increase in the resistant subpopulations. The results of the one-compartment PK/PD models conducted against each strain are shown in Figure 2.

## 3. Discussion

Although infections with the DNS and DNS-VISA phenotypes are not common, they are associated with increased mortality and morbidity, as well as failure with monotherapy therapeutic options [7]. In this study, we have shown that DAL, VAN, and DAP in combination with an anti-staphylococcal cephalosporin have activity against highly resistant MRSA strains, while achieving human-simulated pharmacokinetics. Though the MICs of the strains included in this study varied, we were able to note the decline in the values with the combination therapy, as well as positive synergy and bactericidal activity in the TKAs.

A strength in this present study was the completion of one-compartment in-vitro PK/PD models, simulating human PK parameters and utilizing a glycopeptide, lipopeptide, and lipoglycopetide in combination with CFZ for a 168 h duration against 2 DNS-VISA strains. It has been shown that when used in combination with DAP, beta-lactams have increased synergy and are capable in increasing cationic host defense mediated activity against MRSA as well as preventing the emergence of DAP resistance [13,17,18]. Several studies have shown improved activity with the use of a cell-wall active agent, more specifically CFZ or CPT in addition to DAP through both 96 h PK/PD modelling and TKA [14,19]. Similar to the increased activity observed in these studies; the DAP plus CFZ combination demonstrated rapid and sustained bactericidal activity in each of the in vitro experiments.

In this experiment, we did observe success with the VAN plus CFZ combination; however, in each strain, the combination was met with bacterial regrowth. Other investigators have reported improved activity with VAN plus beta-lactam combinations [16,17]. In the select studies that explore this combination, against VISA strains, each researcher describes that while the combination was significantly better than the monotherapy it did not meet the criteria for therapeutic enhancement [14,15]. The VAN resistance, present in VISA strains, has been associated with the thickening of the bacterial cell wall attributed to an overproduction of loosely cross-linked peptidoglycan preventing the adequate binding of VAN to the cell wall structure [20,21]. It has been described that the presence of the beta-lactam decreases cell wall thickness, allowing for target specific VAN binding [22,23]. Although it has been shown that all VAN-resistant strains undergo this alteration to their cell wall, all adjustments are not equivalent; this can be attributed to the lack or inconsistent activity of the combination contrary to previously mentioned success [24,25]. The presence of VAN-resistant subpopulations has also been recognized as a contributor to VAN failure, and possible failure with the combination of glycopeptides with additional antimicrobials [26]. As both strains were VISA, we did confirm the existence of resistant subpopulations in the D712 and 6913 parent strains. However, we did not observe a shift in the PAP/AUC ratio in the organisms recovered from the VAN plus CFZ models when compared to the PAP analyses of the parent, D712 and 6913 strains. Negating the rationale of an increase in resistant subpopulations attributing to the regrowth observed in the VAN + CFZ combination models.

While the CAMERA-2 study predominately evaluating vancomycin combined with flucloxacillin did not find lower mortality associated with this combination, there was a significantly lower rate of persistent bacteremia. Unfortunately, due to serious concerns for increased nephrotoxicity with this combination the study was terminated early. In addition, because there were too few patients receiving the of combination VAN or DAP plus cefazolin, no meaningful conclusions could be drawn regarding potential benefits from these combinations. Furthermore, none of the clinical isolates evaluated were vancomycin or daptomycin non-susceptible. Our study specifically tested DNS-VISA strains in the PK/PD models, providing data on the impact that the combination of CFZ plus a DAL, VAN or DAP may provide in activity against these phenotypes [27].

Although DAL + CFZ did show a 16 h improvement in time to bactericidal activity, the DAL monotherapy was shown to have one of the highest and more sustained activities against the evaluated strains. Nonetheless, previously published reports have described the evolution of VSSA to VISA following the administration of DAL [28,29]. As DAL and VAN have a similar mechanism of action, it is possible that the two antibiotics bare similar mechanisms of resistance [10]. With that, it has been previously shown that the addition of CFZ to a glycopeptide antibiotic can prevent the emergence of resistance [16]. Based upon this and the positive results observed in our study, the addition of CFZ to DAL may also be a viable therapeutic option to prevent the emergence of glycopeptide or lipoglycopeptide resistant phenotypes.

A potential limitation to this study would be the lack in the completion of de-escalation studies including the combination therapies. Previously, we have shown in vitro the success in de-escalating DAP therapy when used in combination with another antimicrobial, such as CPT or trimethoprim/sulfamethoxazole. It may be possible following the initiation of the combination it may be possible to de-escalate the CFZ therapy [30].

## 4. Materials and Methods

### 4.1. Bacterial Strains

Twenty isolates, of varying *S. aureus* phenotypes including MRSA, hVISA, VISA, DNS, and DNS-VISA obtained from the Anti-infective Research Laboratory (ARL) library were included in the study for susceptibility testing via the broth microdilution method. Four isolates, 2 DNS and 2 DNS-VISA, were selected for TKA. Two isolates, each DNS-VISA, were randomly selected for evaluation in a 168 h one-compartment in vitro PK/PD model.

### 4.2. Antimicrobials and Media

VAN, DAP, and CFZ were purchased through Sigma Chemical Company (St. Louis, MO, USA), while the DAL analytical powder was provided by its manufacturer (Allergan, Parsippany, NJ, USA). In-vitro experiments were completed in standard Mueller Hinton Broth (MHB; Difco, Detroit, MI, USA) supplemented with 25 mg/L calcium and 12.5 mg/L magnesium. In all experiments using DAP, MHB 50 supplemented with Ca^2+^ was utilized. Owing to the tendency of DAL to attach to the plastic involved in the modes of in vitro testing, 0.002% polysorbate 80, commonly referred to as tween (Sigma Chemical Company, St. Louis, MO, USA), was supplemented for all DAL experiments per the CLSI guidelines [31]. To maintain the VISA phenotype of the selected VISA isolates, brain heart infusion agar (BHIA; Difco laboratories, San Jose, CA, USA) was supplemented with VAN (1 mg/L) and utilized in the sub-culturing of the isolates. An MIC of ≥4 mg/L and ≥2 mg/L to VAN and DAP, respectively, was verified via broth microdilution susceptibility testing prior to use in the in-vitro PK/PD models.

### 4.3. Susceptibility Testing

The values of DAL, VAN, DAP and CFZ were determined in duplicate by the broth microdilution (BMD) method, in a 96-well plate, with an inoculum of approximately 10^6^ CFU/mL per the CLSI guidelines [31]. Due to CFZ at half its MIC being greater than the biological peak concentration, in each of the strains evaluated (MICs to each strain were >64 mg/L), DAL, VAN, and DAP were evaluated in combination with CFZ at its fixed biological free peak concertation of 26 mg/L (23). The ATCC 29213 *Staphylococcus aureus* strain was utilized as the quality control strain in the MIC testing. All MIC plates were incubated at 37 °C for 18–24 h prior to being read, in accordance with the CLSI guidelines [31].

### 4.4. In Vitro Time-Kill Analysis

Time-Kill analysis (TKA) was performed on 4 strains, 2 DNS and 2 DNS-VISA, using 24-well tissue culture plates. The experiments were conducted with an inoculum of approximately 10^6^ CFU/mL, with each test being performed in duplicate. TKA was performed on growth control, DAL, VAN, DAP, CFZ, DAL + CFZ, VAN + CFZ, and DAP + CFZ, respectively. DAL, VAN, and DAP were tested at 0.5× MIC for each respective organism in combination with CFZ at its biological free peak concentration, 26 mg/L. The TKAs were incubated in a shaker incubator for 24 h, with 0.1 mL aliquots being removed from each well at 0, 4, 8, and 24 h for analysis. Each sample was serially diluted in 0.9% normal saline to the most appropriate bacterial concentration and then plated using an automated spiral plater. (EasySpiral Pro, Intersciences, Worborn, MA, USA). In an effort to eliminate the DAL carryover in the DAL-containing samples (DAL alone and DAL + CFZ), the centrifugation method was utilized twice as reported previously [32]. Briefly, the samples were centrifuged, following the centrifugation 0.9 mL of the supernatant was removed, and 0.9 ml of normal saline was added in its place. The final sample was then vortexed and serially diluted in normal saline. Similar to the DAL-free samples, the bacterial counts were determined through the spiral plating of the appropriate dilutions using an automatic spiral plater [33] (EasySpiral Pro, Intersciences, Worborn, MA, USA). Plates were incubated at 37 °C, and the colonies were enumerated for 18–24 h following incubation using an automated colony counter (Scan 1200, Interscience Laboratories Inc., Woburn, MA, USA). We have determined these methods to have a reliable lower limit of detection equal to 100 CFU/mL [33]. TKA curves were generated through the plotting of the mean colony-forming units remaining from the experiments at the conclusion of each time point (Graphpad Software, San Diego, CA, USA). A log reduction of ≥2-log_10_ CFU/mL in the combination therapies, from the most active single agent, was considered to be synergistic activity, while the log reduction of ≥3-log_10_ CFU/mL of the combination therapy bacterial count from the initial inoculum was considered to be bactericidal activity.

### 4.5. In Vitro PK/PD Model

Two DNS-VISA strains, 6913 and D712, were tested in an in vitro one-compartment PK/PD model using a 250 mL capacity input and output ports. The model was prefilled with media, and antimicrobials were delivered by bolus injections over a 168 h time period. Prior to the completion of each model, bacterial lawns, created using 0.5 MacFarland of each strain, were grown overnight on TSA plates. The lawns were then suspended and added to each model to obtain a starting inoculum of approximately 10^7^ CFU/mL. Fresh media was continuously supplied using a peristaltic pump at an appropriate rate calculated based upon the half-lives of each antimicrobial. The regimens conducted against each strain included: a growth control, DAL simulating 1500 mg × day 1 ((maximum free-drug concentration in serum, fCmax), 30.1 mg/L; half-life (t_½_) 187.4 h, protein binding 93% [34], VAN simulating 2g every 12 h (q12h) fCmax 36.0 mg/L; t_½_ 6 h [35], protein binding 50%, DAP simulating 10 mg/kg q24h fCmax 14.11 mg/L; t_½_ 8 h, protein binding 94% [36], CFZ simulating 2g q8h fCmax 26.0 mg/L; t_½_ 2.8 h, protein binding 92% [37], DAL 1500 mg × 1 plus CFZ 2g q8h, VAN 2g q12h plus CFZ 2g q8h, and DAP 10 mg/kg q24h plus CFZ 2g q8h). The models were performed in duplicate to ensure the reproducibility of the results. For the combination models, supplemental VAN and DAP were added at appropriate rates to account for the faster half-life of CFZ. DAL was supplemented through a continuous infusion in the DAL plus CFZ combination model.

### 4.6. Pharmacodynamic Analysis

Pharmacodynamic samples were collected in duplicate from each model at the 0, 4, 8, 24, 32, 48, 72, 96, 120, 144, 168 h time points. Samples not containing DAL were serially diluted and plated using an automatic spiral plater; DAL antibiotic carryover was accounted for as described above in TKA methods. Bacteriostatic activity was defined as >2log_10_ CFU/mL decrease from the initial inoculum while bactericidal activity was defined as a >3log_10_ CFU/mL decline from baseline. The graphs for each tested regimen were generated using Prism (GraphPad Software San Diego, CA, USA).

### 4.7. Pharmacokinetic Analysis

Pharmacokinetic samples were collected in duplicate from each model at the 0, 2, 4, 6, 8, 24 in those models that were not completed with DAL. Due to the longer half-life of DAL, PK samples were collected at the 0, 2, 4, 6, 8, 24, 32, 48, 72, 96, 120, 144, and 168 h time points, to verify the attainment of the targeted half-lives. All samples were stored at −80 °C until analysis. DAL concentrations were determined by bioassay using the *Kocuria rhizophilia* ATCC 9341 strain. Blank ¼ in disks were spotted with 20 μL of standard concentration of each sample. Each sample was tested in duplicate by placing the disk on agar plates (grown on antibiotic medium 11) inoculated with 0.5 MacFarland of the standard organism. The assay demonstrated a coefficient variation of less than 11% and 10% for DAL for 10, 20, 30, 40 and 50 mg/L standards. For CFZ, a similar process was used to determine concentrations. However, each sample was tested on agar plates, grown antibiotic medium 5 plates, using the *Bacillus subtilis* ATCC 6633 strain. The assay demonstrated a coefficient variation of less than 10% for 10, 20, 30, 40 and 50 mg/L standards. VAN and DAP samples were analyzed using high-pressure liquid chromatography (HPLC). The DAP PK evaluations were confirmed by a validated high-performance liquid chromatography assay that conformed to the guidelines set forth by the College of American Pathologists [38]. The VAN samples were analyzed using our HPLC methods, as previously described [32]. The HPLC assays demonstrated a coefficient of variation less than 10% for all standards, performed in duplicate. The half-life, area under the curve, and maximum concentration were determines using PK analyst Software (v1.10; MicroMath Scientific Software, Salt Lake City, UT, USA). The trapezoidal method was used to determine the *f*AUC.

### 4.8. Resistance

The emergence of resistance was evaluated at the conclusion of each model by plating 100 μL of each sample onto brain heart infusion agar (BHIA) plates supplemented with either DAL, VAN or DAP at 3× the MIC of each antimicrobial. Plates were examined following a 48-h incubation at 37 °C, the resistant colonies that emerged were evaluated through the broth microdilution method to determine the MICs. If resistance was detected at the conclusion of the model, additional tests were conducted to determine the first occurrence of the resistance.

### 4.9. Population Analysis Profile (PAP) Analysis

The D712 and 6913 parent strains, as well as the organisms recovered at the 168 h time points of the VAN+CFZ combination models against each strain, were tested for the presence of glycopeptide-resistant subpopulations and the increase in resistant subpopulations, respectively. The strains were tested via the modified PAP under the curve AUC method, and the well characterized hVISA strain, Mu3, was used as the control [39]. The starting inoculum was approximately 1 × 10^8^ and was then spiral plated onto brain heart infusion agar plates with varying concentrations of VAN (0, 0.5, 1, 1.5, 2, 4, 8 and 16). An isolate was determined to have resistant subpopulations if the PAP/AUC to Mu3 ratio was ≥0.9.

### 4.10. Statistical Analysis

Changes in the MIC per the MIC susceptibility testing as well as changes in the CFU/mL and time to bactericidal activity in the in vitro PK/PD models were compared by one-way analysis of variance (ANOVA). A *p*-value < 0.05 was be considered significant. All statistical analyses were performed using SPSS Statistics, IBM SPSS software, version 25.0 (IBM Corp., Armonk, NY, USA).

## 5. Conclusions

Through this study in which human pharmacokinetics were achieved, we were able to demonstrate that CFZ given in combination with VAN or DAP was more effective than either agent given as a monotherapy against highly resistant MRSA phenotypes. Additionally, the DAL monotherapy was as active as each combination. These therapies present as potential treatment regimens for serious *S. aureus* infections complicated by glycopeptide, lipopeptide, or lipoglycopeptide resistance. Further research is warranted both in vivo and in vitro to confirm the potential role of DAL, VAN, and DAP plus a beta-lactam in clinical practice.

## Figures and Tables

**Figure 1 antibiotics-09-00696-f001:**
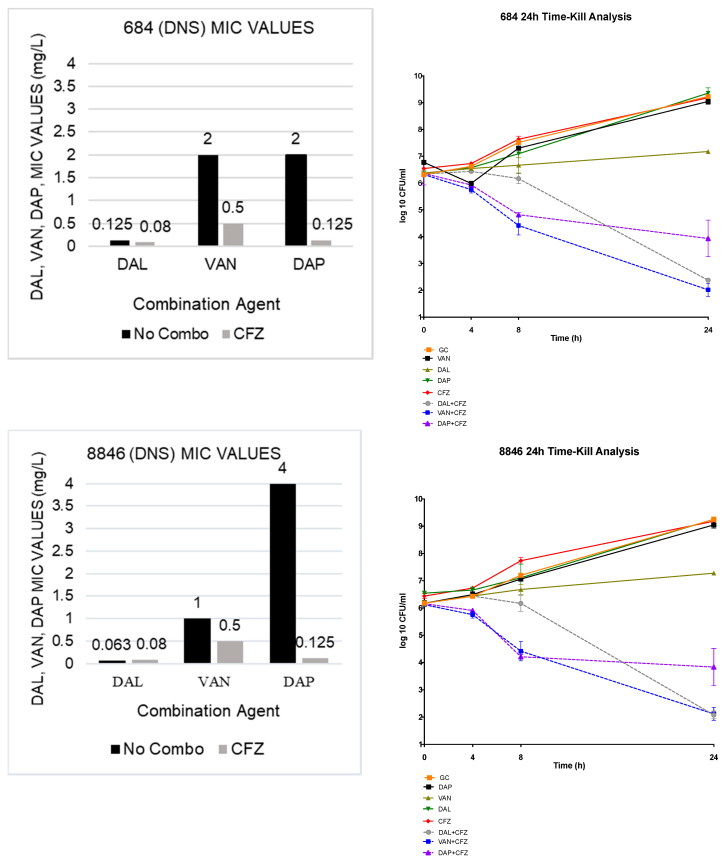
TKA evaluating DAL, VAN and DAP alone and in combination with CFZ on 2 DNS (684 and 8846) and 2 DNS-VISA strains (D712 and 6913).

**Figure 2 antibiotics-09-00696-f002:**
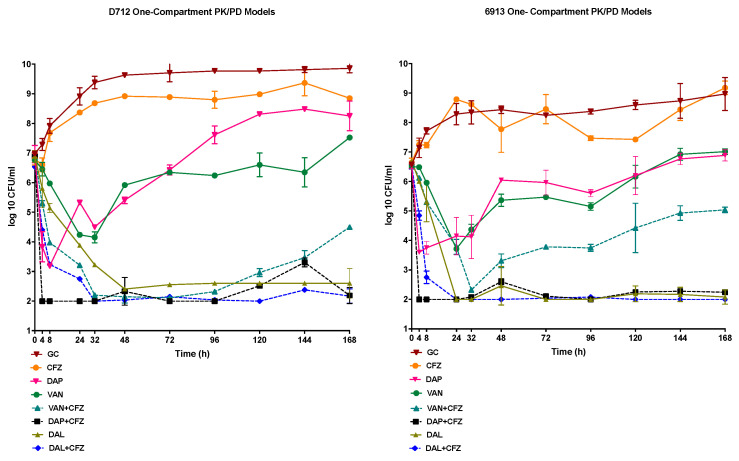
One-compartment PK/PD, 168-h models, completed on the 2 DNS-VISA strains (D712 and 6913), evaluating DAL, VAN, DAP alone and in combination with CFZ.

**Table 1 antibiotics-09-00696-t001:** Summary of MIC values of DAL, VAN, and DAP alone and in combination with CFZ utilizing the Broth Micro-dilution Method (mg/L).

N = 20	DAL	DAL + CFZ	VAN	VAN + CFZ	DAP	DAP + CFZ	CFZ
MRSA	0.016–0.063	0.002–0.008	0.5–2	0.063–0.25	0.063–1	0.0313–0.125	64–≥256
hVISA	0.016–0.125	0.004–0.008	1–2	0.25–0.5	0.063–1	0.063–0.25	64–>256
VISA	0.063–0.125	0.008	4	0.25–1	0.125–1	0.125–0.25	>256
DNS	0.125–0.25	0.008	1–2	0.25–1	2–4	0.125–0.25	>256
DNS-VISA	0.125–0.25	0.008	4	0.25–1	2–4	0.125	>256

MRSA methicillin-resistant *Staphylococcus aureus*, hVISA heteroresistant vancomycin resistant *Staphylococcus aureus*, VISA vancomycin intermediate resistant *Staphylococcus aureus*, DNS daptomycin non-susceptible *Staphylococcus aureus*, DNS-VISA daptomycin non-susceptible vancomycin intermediate *Staphylococcus aureus*, DAL dalbavancin, VAN vancomycin, DAP daptomycin, DAL+CFZ dalbavancin plus cefazolin, VAN+CFZ vancomycin plus cefazolin, DAP+CFZ daptomycin plus cefazolin, CFZ cefazolin.

**Table 2 antibiotics-09-00696-t002:** PK parameters attained and targeted values (represented by parenthesis) in the PK/PD model.

Parameter	DAL	VAN	DAP	CFZ
*f*C_max_ (mg/L)	30.1 ± 0.018 (30.1)	34.9 ± 0.76 (36.0)	13.95 ± 0.04 (14.1)	27.55 ± 0.91 (26.0)
T_1/2_ (h)	184 ± 0.01 (187.40)	6.6 ± 0.03 (6.00)	7.6 ± 0.07 (8.00)	2.4 ± 0.04 (2.3)
AUC (mg × h/mL) or T > MIC (h)	7255± 0.03	324. 66 ± 1.3	192.6 ± 0.09	0.00
*f*AUC/MIC	58,040 (D712) 50,040 (6913)	81 (D712) 40 (6913)	96 (D712) 49 (6913)	234,082 (DAL+CFZ D712); 646 (VAN+CFZ; D712); 1541( DAP+CFZ; D712); 234,082 (DAL+CFZ; 6913) 323 (VAN+CFZ; 6913); 385 (DAP+CFZ; 6913)

Represented in Table 2 are the PK parameters attained in the in-vitro one-compartment models.

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
