# Peer review of "Dalbavancin, Vancomycin and Daptomycin Alone and in Combination with Cefazolin against Resistant Phenotypes of Staphylococcus aureus in a Pharmacokinetic/Pharmacodynamic Model"

_antibiotics, 2020, doi:10.3390/antibiotics9100696_

Round 1

Reviewer 1 Report

The research is very interesting and stimulating for further study.

The authors have shown some evidence: potential treatment regimens for serious human S. aureus infections complicated by glycopeptide, lipopeptide, or lipoglycopeptide resistance.

Minor suggestions are reported below.

  • References in the text: please follow the instructions for authors and use square brackets.
  • Lines 244-247: please refer to a study (bibliography).
  • Lines 328-442: References: please follow the instructions for authors.

Author Response

Reviewer 1:

Minor suggestions are reported below.

  • References in the text: please follow the instructions for authors and use square brackets.

We thank the reviewer for this comment. We have revised the references utilizing the template provided by the journal, the references are not shown in square brackets. As shown in line 36, highlighted in yellow.

  • Lines 244-247: please refer to a study (bibliography).

Thank you to the reviewer for this comment, as these are tools that were used in the laboratory to complete the experiments, there are not studies to be referenced. The references provided are to the company’s website, which offer a photographical description of each specific tool.

  • Lines 328-442: References: please follow the instructions for authors.

We thank the reviewer for this comment. We have revised the references utilizing the template provided by the journal. The updated references are highlighted in yellow.

Reviewer 2 Report

The manuscript by Abdul-Mutakabbir et al. examined the therapeutic effect of adding a beta-lactam to a glycopeptide, lipopeptide, or a lipoglycopeptide and checked the susceptibility of various resistant S. aureus strains. Through a combination of analytical methods and in vitro pharmacodynamic/pharmacokinetic models authors have shown that while Daptomycin(DAP) and vancomycin(VAN) together with cefazolin increase the bactericidal activity, however, dalbavancin (DAV) alone showed equal bacteriocidal activity as in the combination with Cefazolin. The authors concluded that these combinatorial therapies demonstrated in the study as a potential treatment regimen for serious S. aureus infections.

           Overall, the manuscript is well-written except few things (see the comments), interesting to the scientist working with MRSA, the data presented supports the conclusion. The strongest part of this study is the one-compartment in-vitro PK/PD models, simulating human PK parameters and utilizing a glycopeptide, lipopeptide, and lipoglycopetide in combination with CFZ for a 168h duration against  DNS-VISA strains. However, I have a few questions/suggestions which need further attention from authors.

Major issues:

  1. In terms of novelty, I find this manuscript very weak since there are numerous studies done on the same ground and well described in the literature including the antibiotics as reported in this manuscript (Check for details -PMID: 28789974). In the literature (PMID: 28789974), already the combination of VAN and DAP with Beta-lactam have been studies on MRSA. Therefore, What is new observations or conclusions here that are not reported previously? The author should clearly highlight or describe further the novelty of their work either in the introduction or discussion.
  2. Did Author try any other beta-lactam other than Cefazolin such as Oxacillin to check whether the combination therapy is specific to Cefazolin or general towards any Beta-lactam antibiotics against MRSA?
  3. The figure legends should be more descriptive and self-explanatory for both the figures. Currently, the readers have to decipher what the author wants to convey from the figure. The font size should be consistent in all panels of the figure. In figure 1, the graph parts should be improved and enlarged for better visualization.
  4. Line 99 “regrowth being observed at the 24h and 48h time points in DAP and VAN, respectively”. From figure 2 it appears that for D712 one-compartment PK/PD Models DAP treatment after 8 hours there is a jump till 24 hours and then drops for 32 hours and then again increases. The above statement is not clear from what has been shown in the figure. A better explanation/description should be done.

Minor issues:

  1. The author should add the mechanism of action of different antibiotics used in the study to make the rationale of their work clear to the readers.
  2. Line 73 capitalize “staphylococcus”.
  3. Line 122. What is Figure II? It should be figure 2 for consistency.
  4. A better proofread about the capitalization of the letters in the sentence should be done throughout the manuscript.

Author Response

Reviewer 2:

  1. In terms of novelty, I find this manuscript very weak since there are numerous studies done on the same ground and well described in the literature including the antibiotics as reported in this manuscript (Check for details -PMID: 28789974). In the literature (PMID: 28789974), already the combination of VAN and DAP with Beta-lactam have been studies on MRSA. Therefore, what is new observations or conclusions here that are not reported previously? The author should clearly highlight or describe further the novelty of their work either in the introduction or discussion.

We thank the reviewer for the comment, and we have taken into account the previously published research with similar aims investigated. In contrast, our study is the first (to our knowledge) to compare a glycopeptide, lipopeptide, and glycopeptide alone and in combination with an anti-staphylococcal cephalosporin. Further, we evaluated strains with the VISA, VRSA, and DNS phenotypes in the time-kill experiments and one-compartment modeling. There are no reports of dalbavancin, vancomycin, and daptomycin each being tested alone and in combination with cefazolin. Additionally, we saw that the addition of cefazolin improved the activity of each both vancomycin and daptomycin. While activity was not significantly improved with the addition of cefazolin to dalbavancin, we did note the potential utility in the addition of cefazolin should dalbavancin resistant strains arise in the future. We described each of these findings in the discussion section if the manuscript, in lines 146-148, lines 149-150, and lines 180-188, of which is highlighted in the manuscript. We also state our findings in the abstract, and in lines 50-59 (highlighted) of the manuscript we describe our objectives in the completion of the study.

  1. Did the author try any other beta-lactam other than Cefazolin such as Oxacillin to check whether the combination therapy is specific to Cefazolin or general towards any Beta-lactam antibiotics against MRSA?

Thank you to the reviewer for this comment. We did not test any other beta-lactams. We tested only cefazolin in combination with the cell-wall active agents.

  1. The figure legends should be more descriptive and self-explanatory for both the figures. Currently, the readers have to decipher what the author wants to convey from the figure. The font size should be consistent in all panels of the figure. In figure 1, the graph parts should be improved and enlarged for better visualization.

We thank the reviewer for this comment and are in agreement. We have added improved context to each figure legend, this is highlighted in yellow in the manuscript. We have also enlarged the graphs as requested.

  1. Line 99 “regrowth being observed at the 24h and 48h time points in DAP and VAN, respectively”. From figure 2 it appears that for D712 one-compartment PK/PD Models DAP treatment after 8 hours there is a jump till 24 hours and then drops for 32 hours and then again increases. The above statement is not clear from what has been shown in the figure. A better explanation/description should be done.

Thank you to the reviewer for this comment. For clarification, we have adjusted the sentence to read, “Against both the DNS-VISA strains evaluated in the PK/PD model, D712 and 6913, the DAP and VAN monotherapies demonstrated bacteriostatic activity that was not sustained with either antimicrobial, with regrowth being observed at the 48h time points for both DAP and VAN”.

Minor issues:

  1. The author should add the mechanism of action of different antibiotics used in the study to make the rationale of their work clear to the readers.

Thank you to the reviewer for this comment. We have provided the information regarding the class of each antibiotic, the mechanism for each antibiotic has been previously described in the references provided. Specifically, in those provided in the introduction section of the manuscript, lines 34-39 which is highlighted in the text.

  1. Line 73 capitalize “staphylococcus”.

We thank the reviewer or this comment. We have capitalized Staphylococcus.

  1. Line 122. What is Figure II? It should be figure 2 for consistency.

Thank you to the reviewer for this comment, we have changed Figure II to read figure 2, highlighted in the text.

  1. A better proofread about the capitalization of the letters in the sentence should be done throughout the manuscript.

Thank you to the reviewer for this comment. We have done a thorough proofread of the manuscript.

Reviewer 3 Report

The key component of the authors' research seems to be the PK/PD model. However, insufficient detail is given around the method used and discussion of results:

  • In the in vitro PK/PD model, fresh media was supplied at a rate calculated based upon half-lives of each drug. The authors would need to clearly state which half-life values they referred to and show the calculation (equations).
  • Do the half-lives account for other metabolism or elimination that can occur in vivo?
  • How do the in vitro PK/PD model results correlate with in vivo situations and what is the relevance?
  • With what confidence do the authors state that human PK parameters were simulated in the in vitro models (line 136)? What significance or physiological/pharmacological releveance does this have?
  • What does the authors mean by "human pharmacokinetics were achieved" (line 187)?

Author Response

Reviewer 3

In the in vitro PK/PD model, fresh media was supplied at a rate calculated based upon half-lives of each drug. The authors would need to clearly state which half-life values they referred to and show the calculation (equations). Thank you to the reviewer for this comment. We have inserted the ½ lives utilized in the pharmacokinetic area of the manuscript, lies 272-276, highlighted in yellow. We utilized a standardized proportion to derive the flow rate, should the reader need to calculate this the knowledge of the half-life, and duration of the model would be sufficient information. Equation utilized: 0.693/(half-life) x (1hr/60 min) x (24hrs/1 day) x 7 days= flow rate

  • Do the half-lives account for other metabolism or elimination that can occur in vivo?

Thank you to the reviewer for this comment, as the half-lives utilized in the study were taken from in vivo studies, there was an account for additional mechanisms of metabolism or elimination.

  • How do the in vitro PK/PD model results correlate with in vivo situations and what is the relevance?

We thank the reviewer for this comment. The models do correlate with in vivo situations as the pharmacokinetics utilized were extracted from studies in which the antibiotics were used in humans, at the doses utilized in the study. Thus, the in vitro model simulated human pharmacokinetics. This is relevant as it important to present the translational applicability of the in vitro modeling.

  • With what confidence do the authors state that human PK parameters were simulated in the in vitro models (line 136)? What significance or physiological/pharmacological relevance does this have?

Thank you to the reviewer for this comment. We are confident with our statement of achieving human PK parameters. To affirm our confidence, we have completed pharmacokinetic analyses  explained in lines 286-304. The relevance of this includes the translational applicability of the outcomes shown in vitro model to potential in vivo outcomes.

  • What does the authors mean by "human pharmacokinetics were achieved" (line 187)?

We thank the reviewer for this comment. In this line, we are stating that the targeted human PK parameters, as shown in table II, were achieved.

Round 2

Reviewer 2 Report

I am satisfied with the response from the author. Therefore, I do not have any further comments.

Author Response

We thank you the reviewer for their comprehensive comments and review of the manuscript.

Reviewer 3 Report

I appreciate the authors' response regarding the translatability of the PK in their experimental design and interpretation of the results. However, my comments were not only concerning the PK but also the PD aspect of the in vitro-in vivo correlation (or translation).

  • In essence, what is the in vivo relevance of the authors' results in terms of the pharmacodynamic effect (efficacy/potency)? 
  • Are there preceding literature that the authors' approaches are translatable?
  • How do the authors propose the use of their results in translation to in vivo situations?
  • How do the authors expect clinical practice to be affected by their research, if there is to be any?

Author Response

  • In essence, what is the in vivo relevance of the authors' results in terms of the pharmacodynamic effect (efficacy/potency)? 

Thank you to the reviewer for this comment. As the experiments are in vitro, we cannot definitively equate our results to efficacy that would be observed in vivo. Nevertheless, our study did simulate human pharmacokinetics collected in in vivo studies. Utilizing those same PK parameters, we did observe pharmacodynamics similar to those observed in vivo with the use of vancomycin, daptomycin, and dalbavancin against strains with the VISA phenotype. Additionally, we observed similar pharmacodynamics with the use of vancomycin and daptomycin in combination with cefazolin. Thus, we feel that our results are relevant to the potential in vivo results that would be obtained with the antibiotics as the utilized concentrations against the described phenotypes. Based on our own in vitro experiences (Nivedita Singh….Rybak MJ Diagn Micro Infectious Disease), that not only demonstrated potent synergy but prevented the emergence of the VISA phenotype in a well-characterized MRSA strain with a proven proclivity to become VISA after exposure to vancomycin, our medical center (Detroit Medical Center) developed a treatment pathway in 2016 using vancomycin plus cefazolin upfront to improve patient outcomes and reduce the need for escalation to other antibiotics alone and in combination.  Preliminary results were very positive and presented at ID week in 2017(Trinh et al).  We are currently publishing our results (under review) in over 800 patients treated via this pathway. There have been no reports, to our knowledge, that have described the activity of dalbavancin in combination with cefazolin.

Reference:

Singh NB, Yim J, Jahanbakhsh S, Sakoulas G, Rybak MJ. Impact of cefazolin co-administration with vancomycin to reduce the development of vancomycin-intermediate Staphylococcus aureus. Diagnostic microbiology and infectious disease. 2018 Aug 1;91(4):363-70.

Trinh TD, Zasowski EJ, Lagnf AM, Bhatia S, Dhar S, Mynatt R, Pogue JM, Rybak MJ. Combination vancomycin/cefazolin (VAN/CFZ) for methicillin-resistant Staphylococcus aureus (MRSA) bloodstream infections (BSI). InOpen Forum Infectious Diseases 2017 (Vol. 4, No. suppl_1, pp. S281-S281). US: Oxford University Press.

  • Are there preceding literature that the authors' approaches are translatable?

We thank the reviewer for this comment. In a study conducted by Dilworth et al the author discusses the in-vitro success, in studies conducted by other investigators, observed with vancomycin utilized in combination cefazolin. The author states that the success observed in these studies was the basis of their aims in their completed study, which assessed the clinical success of vancomycin in combination with beta-lactams. The in -vitro studies, several of which are included references in our manuscript, included PK/PD modeling, MIC testing, and time-kill analyses (of which was also completed in our study).  Of interest, the included in-vitro studies also used similar modeling methods (one-compartment PK/PD modeling; Hagihara et al) and similar concentrations to the ones used our study. Ultimately the authors observed that similar to the in-vitro studies, those individuals that were treated with the vancomycin plus beta-lactam combination were more likely to experience eradication of MRSA than those that received vancomycin alone. Additionally, in the study entitled, “Daptomycin in Combination with Other Antibiotics for the Treatment of Complicated Methicillin-Resistant Staphylococcus aureus Bacteremia” the authors describe several studies both in vitro and in vivo of Daptomycin alone and in combination with antibiotics, including beta-lactams. In-vitro the studies described did show positive results with the addition of the beta-lactams to daptomycin, including amongst the in-vitro studies mentioned is a study from our group, in which similar methods to this present study were utilized. While in vivo data with daptomycin used in combination with beta-lactams is limited, the authors discuss clinical studies in which the addition of the beta-lactam did decrease daptomycin minimum inhibitory concertation’s, and ultimately resulted in clinical success. Further, our group has shown clinical success with the use of a glycopeptide or lipopeptide plus beta-lactams (Jorgensen et al, Alosaimy et al and Zasowski et al).  Of interest, the antibiotic concentrations utilized in the in vitro and in vivo experiments were similar.

Based upon these mentioned studies, we feel that our approaches were translatable.

To be noted, there is no data, to our knowledge, if dalbavancin utilized in combination with cefazolin.

References:

Dilworth TJ, Ibrahim O, Hall P, Sliwinski J, Walraven C, Mercier RC. β-Lactams enhance vancomycin activity against methicillin-resistant Staphylococcus aureus bacteremia compared to vancomycin alone. Antimicrobial agents and chemotherapy. 2014 Jan 1;58(1):102-9.

Hagihara M, Wiskirchen DE, Kuti JL, Nicolau DP. In vitro pharmacodynamics of vancomycin and cefazolin alone and in combination against methicillin-resistant Staphylococcus aureus. Antimicrobial agents and chemotherapy. 2012 Jan 1;56(1):202-7.

Dhand A, Sakoulas G. Daptomycin in combination with other antibiotics for the treatment of complicated methicillin-resistant Staphylococcus aureus bacteremia. Clinical Therapeutics. 2014 Oct 1;36(10):1303-16.

Jorgensen, S.C., Zasowski, E.J., Trinh, T.D., Lagnf, A.M., Bhatia, S., Sabagha, N., Abdul-Mutakabbir, J.C., Alosaimy, S., Mynatt, R.P., Davis, S.L. and Rybak, M.J., 2020. Daptomycin plus β-lactam combination therapy for methicillin-resistant Staphylococcus aureus bloodstream infections: a retrospective, comparative cohort study. Clinical Infectious Diseases71(1), pp.1-10.

Alosaimy S, Sabagha NL, Lagnf AM, Zasowski EJ, Morrisette T, Jorgensen SC, Trinh TD, Mynatt RP, Rybak MJ. Monotherapy with Vancomycin or Daptomycin versus Combination Therapy with β-Lactams in the Treatment of Methicillin-Resistant Staphylococcus Aureus Bloodstream Infections: A Retrospective Cohort Analysis. Infectious Diseases and Therapy. 2020 Apr 4:1-5.

Zasowski EJ, Trinh TD, Atwan SM, Merzlyakova M, Langf AM, Bhatia S, Rybak MJ. The impact of concomitant empiric cefepime on patient outcomes of methicillin-resistant Staphylococcus aureus bloodstream infections treated with vancomycin. InOpen forum infectious diseases 2019 Apr (Vol. 6, No. 4, p. ofz079). US: Oxford University Press.

  • How do the authors propose the use of their results in translation to in vivo situations?

Thank you to the reviewer for this comment. We propose that in the event that a patient presents with an infection caused by a methicillin-resistant Staphylococcus aureus strain that has decreased susceptibility to vancomycin and/or daptomycin, or in the presence of clinical failure with the use of either of these agents, that practitioners consider the use of dalbavancin alone or either vancomycin, daptomycin, or dalbavancin in combination with cefazolin.  Our study, as well as others have shown that these combinations and dalbavancin alone have shown a greater decline in CFU/ml than vancomycin or daptomycin used alone against the DNS VISA phenotype.

  • How do the authors expect clinical practice to be affected by their research, if there is to be any?

Thank you to the reviewer for this comment. It is our objective to provide viable in vitro results that will offer information on potential antibiotic combinations that should be considered in the presence of decreased susceptibility to vancomycin or daptomycin. With that, we expect that clinicians will take the information that we have presented and in combination with other studies, both in-vivo and in-vitro completed by additional researchers, they will be able to determine an appropriate regimen for patient with an MRSA infection that has not resolved with the standard therapy